# Fabrication of Polypyrrole-Decorated Tungsten Tailing Particles for Reinforcing Flame Retardancy and Ageing Resistance of Intumescent Fire-Resistant Coatings

**DOI:** 10.3390/polym14081540

**Published:** 2022-04-11

**Authors:** Feiyue Wang, Hui Liu, Long Yan

**Affiliations:** Institute of Disaster Prevention Science and Safety Technology, School of Civil Engineering, Central South University, Changsha 410075, China; wfyhn@163.com (F.W.); lhui0421@163.com (H.L.)

**Keywords:** intumescent fire-retardant coatings, tungsten tailing, fire resistance, anti-ageing performance, cooperative effect

## Abstract

Polypyrrole-decorated tungsten tailing particles (PPY-TTF) were prepared via the in situ polymerization of pyrrole in the presence of tungsten tailing particles (TTF), and then carefully characterized by Fourier transform infrared spectroscopy (FTIR), scanning electron microscopy (SEM) and thermogravimetric analysis (TG) analyses. The effect of PPY-TTF on the flame retardancy, smoke suppression property and ageing resistance of intumescent fire-resistant coatings was investigated by a fire protection test, smoke density test and cone calorimeter test. The results show that PPY-TTF exerts excellent cooperative effect on enhancing the flame retardancy and smoke suppression properties of the intumescent fire-retardant coatings, which is ascribed to the formation of more cross-linking structures in the condense phase that enhance the compactness and thermal stability of intumescent char. The cooperative effect of PPY-TTF in the coatings depends on its content, and the coating containing 3 wt% PPY-TTF exhibits the best cooperative effect among the samples, showing a 10.7% reduction in mass loss and 35.4% reduction in flame-spread rating compared to that with 3% TTF. The accelerated ageing test shows that the presence of PPY-TTF greatly slows down the blistering and powdering phenomenon of the coatings, thus endowing the coating with the super durability of fire resistance and smoke suppression property. This work provides a new strategy for the resource utilization of tungsten tailing in the field of flame-retardant materials.

## 1. Introduction

Intumescent fire-retardant coatings are widely used in high-rise buildings, steel structures, ancient buildings, railway stations, airport terminals and other buildings for the advantages of good finish, small coating thickness, being lightweight, and excellent fire protection performance of the components [1,2,3,4], which are considered to be one of the most effective and economical materials to reduce fire hazards on buildings [5,6]. Intumescent fire-retardant coatings generally compose of an intumescent fire-retardant system, binders, synergists, adjuvants and additives, which interact to forge an expanding char layer during combustion that effectively delays the spread of flame and preventing the formation of fire, thus guaranteeing the structural integrity of buildings and minimizing the fire risk [7,8]. Although a large number of studies have concentrated on the enhancement of fire resistance, smoke suppression and reduction in the thickness of intumescent fire-retardant coatings [4,9,10,11], which neglected the ageing degradation of intumescent fire-retardant coatings under the impacts of environmental factors such as certain humidity, UV radiation, temperature and oxygen.

The use of multifunctional additive is an important strategy to enhance the flame-retardant and smoke suppression efficiencies of intumescent fire-retardant materials. Numerous efforts have investigated the effects of additive on fire resistance, smoke suppression and ageing resistance of materials, such as SiO_2_, bio-fillers, kaolinite, graphene, carbon nanotubes and carbon black [3,6,7,12]. TTF enriches in SiO_2_, Al_2_O_3_, CaO, Fe_2_O_3_ and other components, which has high chemical stability and small size of particles. However, TTF is easily agglomerated and migrated on the surface of coatings in actual application. To overcome these limitations, many studies have been concentrated on the fabrication of hybrid materials, such as organic space-stabilized polypyrrole latexes and composites with metal oxide or silica particles [13]. Polypyrrole (PPY) has attracted extensive research in supercapacitors, DNA extraction, microwave absorption and contaminant removal, owing to its superior conductivity, easier synthesis, excellent biocompatibility, high thermal stability and non-toxicity [14,15,16,17,18]. The preparation and characterization of silica–polypyrrole nanocomposite colloids using silica particles have been reported by Armes et al. [19]. Among them, silica/PPY nanocomposites have a low cost, high surface area and easier dispersion, which can improve the wear resistance, UV resistance, barrier property, mechanical performance and thermal stability of organic coatings [20]. Copolymerization can be an opportunity to amend the physical properties and avoid the drawbacks of individual polymers, and several copolymer composites may demonstrate better behavior in many applications [21]. Microencapsulation is an important method of enhancing the compatibility of fillers, and the resulting polypyrrole-decorated tungsten tailing particles (PPY-TTF) are expected to have better dispersion in intumescent fire-retardant coatings via the in situ polymerization of pyrrole, thus obtaining better fire resistance and anti-ageing properties.

Herein, a preparation method for PPY-TTF was proposed via the in situ polymerization of pyrrole in the presence of TTF particles. Five intumescent fire-retardant coatings were prepared with an intumescent fire-retardant system, waterborne epoxy resin and PPY-TTF. The effect of PPY-TTF on the combustion performance of intumescent fire-retardant coatings was analyzed by various analytical methods, and the flame-retardant and smoke suppression mechanisms of PPY-TTF were also proposed.

## 2. Experimental Section

### 2.1. Materials

TTF was obtained from Hunan Anhua Xiangan Tungsten Co., Ltd., Hunan, China. Waterborne epoxy hardener (ERC 2610, amine value:180 ± 40 mg KOH/g) and waterborne epoxy resin (ERE 2581, solid content: 52 ± 1%) were supplied from Ruicheng Selected New Materials Technology Co., Ltd., Guangdong, China. Pyrrole (C_4_H_5_N, purity: 99%) and FeCl_3_·6H_2_O (purity: 99.0%) were offered by Shanghai Aladdin Biochemical Technology Co., Ltd., Shanghai, China. The pentaerythritol (PER, purity: 99.5%), ammonium polyphosphate (APP, water solubility ≤ 0.04%, purity: 99.5%) and melamine (MEL, purity ≥ 99.5%) were available from Hangzhou JLS Flame Retardant Chemical Co., Ltd., Hangzhou, China. The acrylate copolymer carboxylate used as a dispersant was supplied by Qingdao Xingguo Coatings Co., Ltd., Qingdao, China. The non-silicon defoamer was supplied by Qingdao Xingguo Coatings Co., Ltd., Qingdao, China.

### 2.2. Preparation of Polypyrrole-Decorated Tungsten Tailing Filler

Firstly, 30 g TTF was added to 400 mL mixture of anhydrous ethanol and water (2:8) and ultrasonicated for 0.5 h. One milliliter pyrrole was dropped into the solution and dispersed ultrasonically for 10 min. Owing to the charge interaction between the pyrrole and TTF, the pyrrole monomers in solution migrated to the surface of TTF particles. Then, 7.99 g FeCl_3_·6H_2_O was mixed with 100 mL water, stirred to dissolve and sonicated to disperse. The FeCl_3_·6H_2_O solution was slowly dropped into the above solution and the reaction occurred at low temperature, with stirring for 24 h to polymerize the pyrrole monomers on the surface of TTF particles. The PPY-TTF was obtained by extracting and drying at 60 °C for 6 h. The synthetic route of PPY-TTF is shown in Figure 1.

### 2.3. Preparation of the Intumescent Fire-Retardant Coatings

The intumescent flame retardant (IFR) was obtained by mixing APP, PER and MEL with a mass ratio of 3:1.5:1, which was combined with PPY-TTF and deionized water according to the formula in Table 1 and then stirred at 1000 r/min for 20 min. Then, waterborne epoxy resin, defoamer and dispersant were put into the slurry stirring at 500 r/min for 20 min. Finally, the waterborne epoxy hardener was added and stirred at 500 r/min for 20 min to obtain the intumescent fire-retardant coatings. The intumescent fire-retardant coatings with different amounts of PPY-TTF are named IFRC_0_-IFRC_5_, and the specific test formulations are shown in Table 1.

### 2.4. Characterization

#### 2.4.1. Fourier Transform Infrared Spectroscopy

Fourier transform infrared spectroscopy (FTIR) spectra were characterized by an iCAN9 FTIR spectrometer using KBr pellets (Tianjin Energy Spectrum Technology Co., Ltd., Tianjing, China).

#### 2.4.2. Scanning Electron Microscopy

The MIRA 3 LMU scanning electron microscopy (Tescan, Brno, Czech Republic) was used to characterize the micromorphology of the samples at a voltage of 20.0 kV.

#### 2.4.3. X-ray Diffraction

X-ray diffraction (XRD) pattern was characterized by an Advance D8 XRD diffractometer at a range of 3 to 70°, a Cu-Ka radiation and a rate of 5°/min (Bruker, Fällanden, Switzerland).

#### 2.4.4. X-ray Fluorescence Spectrometer

X-ray fluorescence spectrometer (XRF) analysis was performed with a ZSX Primus II X-ray Fluorescence Spectrometer (Rigaku, Japan).

#### 2.4.5. Smoke Density Test

Smoke density test was performed on a PX-07-008 smoke density tester for building materials (Phoenix Quality Inspection Instruments Co., Ltd., Suzhou, China).

#### 2.4.6. Fire Protection Tests

Fire protection properties of the samples were assessed by the big panel method, cabinet method, tunnel method tests according to GB12441-2018 standard procedure. The big panel method test was conducted on a MT-X multiplex temperature recorder (Shenzhen Shenhwa Technology Co., Ltd., Shenzhen, China) to obtain the backside-temperature curves of coating samples.

Cabinet method test was carried out on an XSF-1 apparatus, the weight loss (the difference in the sample mass before and after the test), char index and intumescent factor (the ratio of the thickness of the coatings after and before test) of the samples were determined. The char index was calculated by Formula (1):(1)$$Char\;index=\frac{\sum_{i=1}^{n}(a_{i}b_{i}h_{i})}{n}$$
where *a_i_* is defined as the char length (cm), *b_i_* is defined as the char width (cm), *h_i_* is defined as the char depth (cm) and *n* is the number of samples.

Tunnel method test was conducted on a SDF-2-type 2-foot flame tunnel instrument (Jiangning Analysis Instrument Company, Nanjing, China). The flame spread over the coating surface of the samples was evaluated when ignited under controlled conditions in a small tunnel, and the flame-spread rating of the samples was calculated by Formula (2):(2)$$FRS=\frac{L_{s}-L_{a}}{L_{r}-L_{a}}$$
where *L_s_* is the mean of five flame advance readings of samples (mm), *L_a_* is the mean of five flame advance readings of asbestos board (mm) and *L_r_* is the mean of five flame advance readings of oak board (mm).

#### 2.4.7. Adhesion Classification Test

The adhesion classification test was carried out on a QFH-HD600 adhesion tester to test the coating adhesion (Changzhou Edex Instruments Co., Ltd., Changzhou, China) on the basis of ASTM D 3359-09.

#### 2.4.8. Pencil Hardness Test

The pencil hardness test was performed on a QHQ-A portable pencil scratch tester in accordance with ISO 15184-2012. The lead of a pencil was vertically erected and polished on sandpaper for a flat and round cross-sectioned end. The pencil hardness of the coating was tested progressively by placing a polished pencil on the coating at an angle of 45° under a load of 750 g.

#### 2.4.9. Cone Calorimeter Test

The cone calorimeter test was used to characterize the heat release rate (HRR) and total heat release (THR) curves of coatings with an external heat flux of 50 kW/m^2^.

#### 2.4.10. Thermogravimetric Analysis

TG analysis was carried out on a TGA/SOTA 851 thermogravimetric instrument (Mettretoli Instruments Co., Ltd., Zurich, Switzerland) from 30 °C to 800 °C at a heating rate of 10 °C/min under a nitrogen atmosphere. The theoretical char residue (*W*_theo_) of IFRC_0_-IFRC_5_ samples was calculated using Formula (3):(3)$$W_{\mathrm{theo}}\left(t\right)=\overset{n}{\underset{i=1}{\sum }}\chi_{i}W_{i}\left(t\right)$$
where $W_{i}\left(t\right)$ is the amount of residual char for *i* at *t* °C; $\chi_{i}$ is the percentage of *i*, %.

#### 2.4.11. Accelerated Ageing Test

The accelerated ageing test was performed on an UV-accelerated ageing tester (Shi Haoran Machinery Equipment Factory, Dongguan, China) according to ASTM G154-2006. One ageing cycle included 12 h, during which, the condensation time was 4 h at 50 ± 3 °C, and the UV exposure was 8 h at 60 ± 3 °C with an irradiation of 0.76 W/(m^2^·nm). The samples were exposed to accelerated ageing tests for 2, 6 and 11 cycles, accompanying with the change of sample position every 12 h.

## 3. Results and Discussion

### 3.1. Morphology and Composition of TTF and PPY-TTF

The XRD patterns of TTF are presented in Figure 2. From Figure 2, the XRD patterns of TTF samples are generally consistent with the standard patterns of quartz (SiO_2_) (PDF85-0794). The major diffraction peaks located at 21.0°, 26.8°, 36.7°, 39.6°, 40.5°, 42.7°, 46.0°, 50.3°, 55.1°, 55.5°, 60.1°, 64.2°, 67.9° and 68.3° are attributed to (100), (011), (110), (102), (111), (200), (201), (112), (022), (013), (121), (113), (122) and (203) phases of SiO_2_, indicating that the main component of TTF is SiO_2_. XRF is used for analyzing the elemental composition of TTF, and the results are illustrated in Table 2. From Table 2, it is clearly seen that the main component of TTF is SiO_2_, which is in agreement with the XRD result.

Figure 3 exhibits the SEM images and EDS maps of TTF and PPY-TTF. From Figure 3a, it can be seen that the TTF sample shows irregular blocky structures with a size range of 10 and 50 μm. Compared to TTF, the size of PPY-TTF is obviously reduced, showing a smaller agglomeration phenomenon. From EDS maps, C and N elements from polypyrrole are homogeneously distributed on the surface of TTF, indicating that PPY-TTF was prepared successfully.

Figure 4 shows the FTIR spectra of TTF and PPY-TTF. In the spectrum of TTF, the characteristic peaks at 1005, 777 and 459 cm^−1^ are attributed to the symmetric and antisymmetric stretching vibration peaks of Si–O–Si [22,23]. As for PPY-TTF, the new peaks of the C–H stretching vibration (2928, 2859 cm^−1^), C=C stretching vibration (1631 cm^−1^) and C–N and N–H stretching vibration (1551 cm^−1^) are observed [23,24,25], further confirming polypyrrole was successfully decorated on the surface of TTF particles.

Figure 5 shows the TG and differential thermo-gravimetric (DTG) curves of TTF and PPY-TTF. From the TG and DTG curves of TTF, it can be seen that the pyrolysis process of TTF is mainly from 420 °C to 800 °C concomitant with a strong DTG peak and a residual weight of 96.0% at 800 °C. Meanwhile, the pyrolysis process of PPY-TTF is mainly divided into four stages. The first stage corresponds to the temperature interval of 26–200 °C, which is due to physics-absorbing water molecules, oligomer molecules and volatile impurities. The second stage at 200–430 °C is attributed to the degradation of low molecular-weight polymer chains. The third stage at 430–560 °C is the decomposition stage of polypyrrole. The fourth stage at 560–800 °C is the decomposition stage of TTF and polypyrrole, remaining an 81.8% residue at 800 °C.

### 3.2. Fire Protection Tests

The results of the tunnel method and cabinet method tests are given in Table 3. The mass loss, charring index, flame-spread rating and intumescent factor of IFRC_0_ samples are 3.7 g, 34.9 cm^3^, 20.5 and 25.0, respectively, while the addition of PPY-TTF can significantly enhance the fire resistance of the intumescent fire-retardant coatings. In particular, IFRC_3_ presents the best heat-insulation performance, with a 32.9% reduction in weight loss, 46.7% reduction in charring index, 74.3% reduction in flame-spread rating and 80.0% increase in intumescent factor compared with IFRC_0_. The results indicate that the appropriate amount of PPY-TTF obtained via the in situ polymerization of pyrrole can effectively strengthens the fire protection of coatings. In addition, an excessive amount of PPY-TTF may suppress the expansion and carbonization of the coatings, thus diminishing its cooperative fire resistance in intumescent fire-retardant coatings.

Figure 6 shows the backside temperature curves of the substrates coated with IFRC_0_-IFRC_4_ coatings. As demonstrated in Figure 6, the backside temperature of the IFRC_0_ sample without PPY-TTF rises rapidly, with a fire resistance time of 800 s, corresponding to a poor heat-insulation performance. The backside temperature of IFRC_1_-IFRC_4_ samples containing PPY-TTF has a slower rise and becomes stable at about 500 s. The equilibrium backside temperatures of the samples at 900 s are 180.0, 167.7, 142.3 and 156.2 °C, respectively, indicating that the presence of PPY-TTF enhances the fire protection of the coatings. Among them, the IFRC_3_ sample presents the best fire-retardant effect, which is consistent with the results in Table 3.

Figure 7 exhibits digital photographs of the char layers after the smoke density test, where the IFRC_0_ char presents the smallest expansion height and the poorest surface structure, thus showing the worst fire-retardant properties. After the addition of PPY-TTF, the char layer heights of IFRC_1_–IFRC_4_ are 8.5 mm, 11.5 mm, 14.5 mm and 12.5 mm, respectively, revealing that the presence of PPY-TTF enhances the carbonization and expansion process of the coatings. In particular, the IFRC_3_ sample exhibits the highest intumescent factor and the densest char. From the SEM images of Figure 8, it can be found that the introduction of PPY-TTF facilitates the formation of a denser and more continuous char layer structure without obvious holes or other defects that effectively blocks the transfer of combustible materials and heat. From the EDS maps of Figure 8, the higher content of P and Si elements in IFRC_3_ are observed, which indicate that the presence of PPY-TTF is favorable to the formation of more phosphorus-rich and silicon-rich cross-linking structures that enhance the heat-insulation performance of the char layer. Moreover, the char layer of IFRC_3_ has a higher C/O mass ratio compared to that of IFRC_0_, resulting in a good oxidation resistance and fire resistance [7].

### 3.3. Cone Calorimeter Test

The THR and HRR curves of the IFRC_0_–IFRC_4_ samples are presented in Figure 9 and Figure 10, respectively. As depicted in the figures, the samples illustrate two peaks due to the decomposition of the char layer and the plywood. The first peak heat release rate (PHRR1) of IFRC_0_ is 108.5 kW/m^2^ appeared at 21 s. With the addition of PPY-TTF, both THR and PHRR values of IFRC_1_–IFRC_4_ coatings are significantly reduced. Compared with IFRC_0_, the THR and PHRR1 values are reduced by 5.9% and 12.1% for IFRC_1_, 13.5% and 18.5% for IFRC_2_, 21.6% and 31.2% for IFRC_3_ and 10.9% and 27.2% for IFRC_4_, suggesting that the presence of PPY-TTF can effectively improve flame retardancy of the coating. Moreover, the coatings containing PPY-TTF show lower second peak heat release rate (PHRR2) compared to the coating without PPY-TTF, indicating better flame inhibition effect on wood substrates. In conclusion, the coatings containing PPY-TTF can significantly reduce the THR and HRR of coating samples, among which, the IFRC_3_ has the best performance. This phenomenon is mainly attributed to the fact that PPY-TTF can effectively promote the char formation of the coating and strengthen the structure of the char layer, thus effectively delaying the transfer of heat and mass between the flame and the substrates. However, the cooperative effect of PPY-TTF in the coatings depends on the amount of PPY-TTF. When the amount of PPY-TTF exceeds 3 wt%, the cooperative fire-retardant effect will be weakened, which is ascribed to an excessive content of PPY-TTF may solidify the molten char layer that inhibits the char formation process [20].

### 3.4. Smoke Density Test

Figure 11 presents the light absorption curves and smoke density rating (SDR) values of the wood substrates coated with intumescent fire-resistant coatings. The maximum light absorption values of IFRC_0_–IFRC_4_ are 59.0%, 56.2%, 52.2%, 40.6% and 47.7%, and the smoke density rating values are 33.3%, 31.3%, 24.4%, 22.5% and 23.6%, respectively. The addition of PPY-TTF can effectively reduce the smoke density rating values and light absorption rate of coatings, among which the IFRC_3_ sample shows the optimal smoke suppression performance. The super smoke suppression effect of PPY-TTF in the coating is ascribed to the formation of a dense and compact char against the release of pyrolysis products during combustion.

### 3.5. Thermal Stability Analysis

The TG and DTG curves of coatings under a nitrogen atmosphere are depicted in Figure 12. As seen in Figure 12, the decomposition process of the coating is mainly divided into four stages in the temperature range of 100–300 °C, 300–430 °C, 430–580 °C and 580–800 °C, respectively. The first stage is mainly the stage of dehydration, volatilization for small molecules and low temperature decomposition of IFR with lower mass loss. As the temperature continues to rise, the second stage appears a strong DTG peak accompanied with a higher mass loss, which is mainly caused by the decomposition of the cured epoxy resin and APP, PER and MEL. In this stage, APP decomposes to generate polyphosphoric acid and phosphoric acid derivatives that promote the esterification of PER into char, while polyphosphoric acid dehydrates to form a cross-linking structure. The NH_3_ and H_2_O released from MEL and APP in this process can promote the expansion of the char layer. In the third stage, the formed cross-linking structures and polypyrrole are degraded at a large scale. The fourth stage is ascribed to the decomposition of unstable char layer at a high temperature, accompanying with a mass loss of about 1.5%.

Table 4 illustrates the relevant thermal decomposition parameters.As seen in Table 4, T_0_ and T_m_ values of IFRC_0_ sample occur at 223.1 °C and 362.9 °C, respectively, and the residual weight at 800 °C is 26.7%. With the addition of PPY-TTF, the coatings show lower T_0_, T_m_ and mass loss, suggesting a reduction in the release of pyrolysis products and an increase in the amount of residual char. Generally, the higher the Δ*W* value of the sample, the stronger the interaction between the components. After adding PPY-TTF, the *W*_exp_ of the specimens is significantly higher than their *W*_theo_. Among them, IFRC_3_ shows the maximum Δ*W* value of 15.9% corresponding to the best char-forming efficiency. The results indicate that PPY-TTF can effectively improve the structure of the char layer and encourage the cross-linking reaction of degradation products to strengthen the thermal stability and char-forming ability of coatings, thus exhibiting a super cooperative effect.

### 3.6. Accelerated Ageing Test

Figure 13 presents the morphologies of the IFRC_0_ and IFRC_3_ samples before and after the ageing treatment. With the increased ageing time, the obvious blistering, powdering and yellowing phenomena appear on the surface of the IFRC_0_ coating. The powdering phenomenon is caused by the precipitation and decomposition of IFR under the influence of UV irradiation and hydrothermal conditions, whereas the blistering phenomenon mainly results from the reduction of the adhesion of the coatings. Figure 14 exhibits the adhesion classification and pencil hardness of IFRC_0_ and IFRC_3_ coatings after the accelerated ageing treatment. As shown in Figure 14, the adhesion classification and pencil hardness of the coatings gradually weaken with the increase of ageing cycles. However, the IFRC_3_ coating has a better performance than that of IFRC_0_ under the same ageing treatment. This may be explained by the reason that the introduction of PPY-TTF can effectively strengthen the hardness, adhesion and shielding effect of the coating and weaken the ageing degradation of the coatings, thus exhibiting less bubbles and the precipitation of flame retardants, as seen in Figure 13. Therefore, the addition of PPY-TTF can effectively enhance the durability of the coating and slow down the blistering, powdering and yellowing of the coatings.

The smoke emission characteristics of IFRC_0_ and IFRC_3_ coatings before and after ageing treatment are presented in Figure 15. As depicted in Figure 15, the maximum light absorption rating and SDR values of IFRC_0_ and IFRC_3_ samples gradually increase with the growth of the ageing cycle, reflecting the increase of smoke production. Compared with IFRC_0_, the IFRC_3_ sample still expresses a better cooperative smoke-suppression effect after the same accelerated ageing treatment, which is ascribed to the enhanced integrity of the coatings after introduction of PPY-TTF under ageing treatment. It can be concluded that the presence of PPY-TTF can strengthen the durability of smoke-suppression effect of intumescent fire-retardant coatings.

The fire resistance of IFRC_0_ and IFRC_3_ samples after the ageing treatment are given in Figure 16. As shown in Figure 16, IFRC_0_ and IFRC_3_ have a reduction in fire-retardant time with the increase of ageing time, indicating that the degradation of fire protection performance. Compared with IFRC_0_, the ageing process has a less negative effect on the fire resistance of IFRC_3_ coating. The reason is that the addition of PPY-TTF is beneficial to weaken the migration of flame retardants and morphology deterioration of the coatings, thus imparting the coatings with a long-term durability of fire resistance.

Intumescent fire-retardant coatings are susceptible to photo-oxidation and thermal oxidation in daily use, and the evolution processes of coatings under ageing conditions are monitored by FTIR analysis. Figure 17 presents the FTIR spectra of IFRC_0_ and IFRC_3_ coatings, and Table 5 gives the functional groups of IFRC_0_ and IFRC_3_ samples after different ageing cycles. As observed in Figure 17 and Table 5, the absorption peaks of –NH_2_ (3470, 3419, 1439 cm^−1^), C=N (1654 cm^−1^), N–H (1552, 3134 cm^−1^), P=O (1249 cm^−1^) and C–O (1016 cm^−1^) groups are obviously strengthened with the increase in the ageing cycle [24,26,27,28,29,30,31,32], indicating that flame retardant migrates to the surface of the coating under the influence of ageing factors such as irradiation, oxygen, humidity and temperature. However, after eleven ageing cycles, the intensity of the absorption peaks for –NH_2_, C=N, N–H, P=O, and C–O functional groups decrease remarkably and the –NH_2_ and N–H groups disappear, showing that the migration, hydrolization and oxidation of APP, PER and MEL under ageing conditions. Compared with IFRC_0_, the IFRC_3_ coating has stronger absorption vibration peaks for the main functional groups under the same ageing treatment, corresponding to a weaker ageing degradation. This result shows that PPY-TTF can strengthen the structural stability of the coatings and reduce the migration, hydrolization and oxidation of the coatings during ageing conditions, thus endowing the coatings with a better ageing resistance.

### 3.7. Flame Retardant and Smoke-Suppression Mechanisms

To further study the effect of PPY-TTF on the flame retardancy and smoke suppression properties of intumescent fire-retardant coatings, the combustion processes of the cured epoxy resin, IFRC_0_ and IFRC_3_ samples are analyzed using a cone calorimeter, and the results are presented in Figure 18. As shown in Figure 18, a molten layer is formed at about 60 s on the surface of the coating at a radiation flux of 50 kW/m^2^. As the temperature increases, the components of the coatings interact to form an intumescent char layer, which suppresses the further decomposition of the coatings. With the increase in burning time, the formed char layers start to decompose and weaken the barrier effect on the inner coating. The cured epoxy resin starts to burn at 125 s, while the addition of IFR delays the ignition time of the coating to 372 s. More importantly, the IFRC_3_ char maintains its structural integrity at 900 s, which means that the incorporation of PPY-TTF can effectively strengthen the fire resistance of the coatings and provide better protection for the substrate.

The FTIR spectra and digital photos of IFRC_0_ and IFRC_3_ after different treating temperatures are depicted in Figure 19 and Table 6. When the temperature reaches 300 °C, the –NH_2_ (3469, 3419 cm^−1^), N–H (1552 cm^−1^), PO_3_^2−^ (1129 cm^−1^), C–O (1015 cm^−1^) and P–O–P (669, 873 cm^−1^) groups disappear and the P–O–C group appears at 1043 cm^−1^ [23,26,27,28,29,30,31,32,33,34], indicating the decomposition of APP, PER, MEL and epoxy resin at a low temperature. Besides, the decomposition temperature of the N–H group in the IFRC_3_ coating is 100 °C lower than that of the IFRC_0_ sample due to the earlier char formation of the coating. As the temperature continues to increase, the pictures reveal that the components of the intumescent fire-retardant coatings interact to form an intumescent char layer, and the main functional group peaks of the coatings in the corresponding FTIR spectra basically disappear. When the temperature reaches 800 °C, the char residue of the IFRC_3_ sample appears stronger stretching vibration peaks of P=O (1324 cm^−1^), C–O–C (1139 cm^−1^), P–O–C (921 cm^−1^) and aromatic C–H (739 cm^−1^) groups than those of IFRC_0_, indicating that the presence of PPY-TTF causes the char residue to form more phosphorus-rich cross-linking structures and aromatic structures, thus improving the heat insulation and thermal stability of the char layer [35,36,37,38]. This result is consistent with the higher residual weight of IFRC_3_ sample in TG analysis.

During the combustion process, APP is heated and decomposed to release metaphosphate, phosphate, inorganic acid and non-combustible gas, among which inorganic acid can encourage the esterification of PER into char to form a molten layer. At this stage, the introduction of PPY-TTF promotes the production of more cross-linking structures in the condensed phase. At the same time, MEL will decompose and cyclize into triazine compounds, while a large amount of non-combustible gas is released to induce the expansion of the char layer, diluting the fuel gas and weakening the burning intensity. The PPY-TTF will promote the formation of a smoother and denser protective char layer that delays the transfer of heat and mass between the fire and char layer, thus effectively suppressing the further decomposition of the coating. With the rise in temperature, the PPY-TTF contributes to the cross-linking reaction of degradation products that generate more cross-linking and aromatic structures in the condensed phase, which enhance the thermal stability and char-forming properties of coatings. However, the positive effect of PPY-TTF in the coatings is depended on its content. An excessive content of PPY-TTF may inhibit the char formation and decrease the expansion rating of the char layer, thus causing a reduction in the cooperative effect on the fire resistance and smoke suppression performance of intumescent fire-retardant coatings.

## 4. Conclusions

In this work, a preparation method for PPY-TTF was proposed via the in situ polymerization of pyrrole on the surface of tungsten tailing fillers, and the structures and properties of synthesized particles were characterized in detail by combining SEM-EDS, XRD, FTIR, XRF and TG analyses. Then, the PPY-TTF was applied to intumescent fire-retardant coatings as an adjuvant, and the effect of PPY-TTF on the fire resistance and anti-ageing properties of intumescent fire-retardant coatings was investigated by different analytical methods. The results reveal that the presence of PPY-TTF enhances the fire resistance, thermal stability, char formation and smoke suppression properties of intumescent fire-retardant coatings, exhibiting super cooperative flame-retardant and smoke suppression effects. The cooperative effect of PPY-TTF in intumescent coatings is ascribed to the formation of more cross-linking and aromatic structures in the condensed phase that enhance the barrier effect of char, as supported by digital photos and SEM images. However, an excessive content of PPY-TTF will weaken the char-forming ability of the coatings, thus diminishing the excellent cooperative efficiency. In particular, the IFRC_3_ sample containing 3 wt% PPY-TTF presents the best fire resistance among all samples, and has a 74.3% reduction in flame-spread rating, 30.7% reduction in total heat release, 32.9% reduction in mass loss and 32.4% reduction in smoke density rating value compared with IFRC_0_. The TG analysis suggests that PPY-TTF can strengthen the char-forming ability of intumescent fire-retardant coatings, and the residual weights of IFRC_0_, IFRC_1_, IFRC_3_, IFRC_4_ and IFRC_5_ at 800 °C are 26.7%, 31.6%, 34.9%, 33.5% and 30.6%, respectively. The accelerated ageing test demonstrates that an appropriate amount of PPY-TTF can improve the shielding effect and structural stability of the coatings that effectively slow down the blistering and powdering phenomenon of the coatings, thus achieving a long-term durability of the fire resistance and smoke suppression properties of the coatings. In summary, PPY-TTF provides a new strategy to utilize tungsten tailing in the fields of flame-retardant materials.

## Figures and Tables

**Figure 1 polymers-14-01540-f001:**
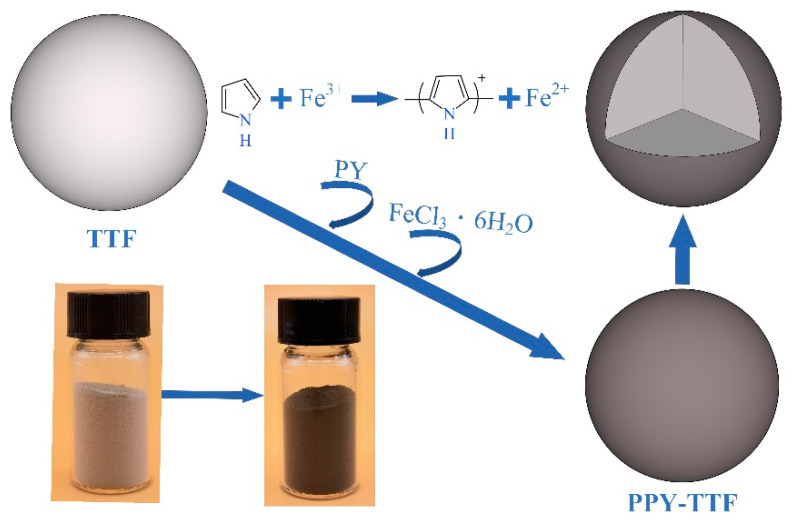
Synthetic route of PPY-TTF.

**Figure 2 polymers-14-01540-f002:**
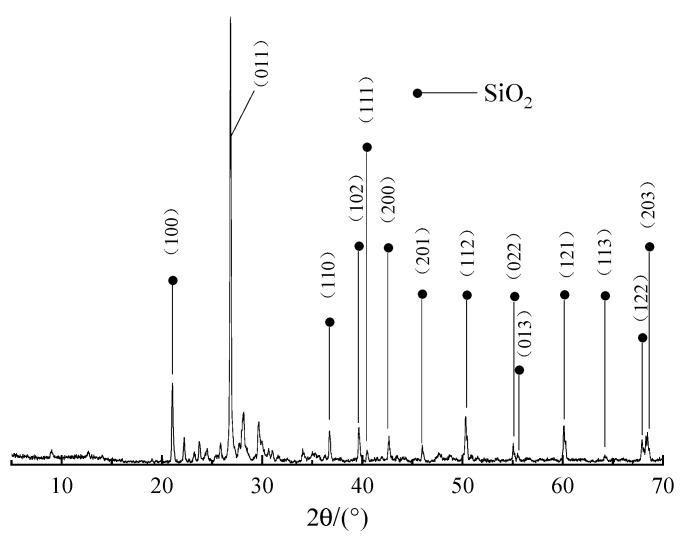
XRD patterns of TTF.

**Figure 3 polymers-14-01540-f003:**
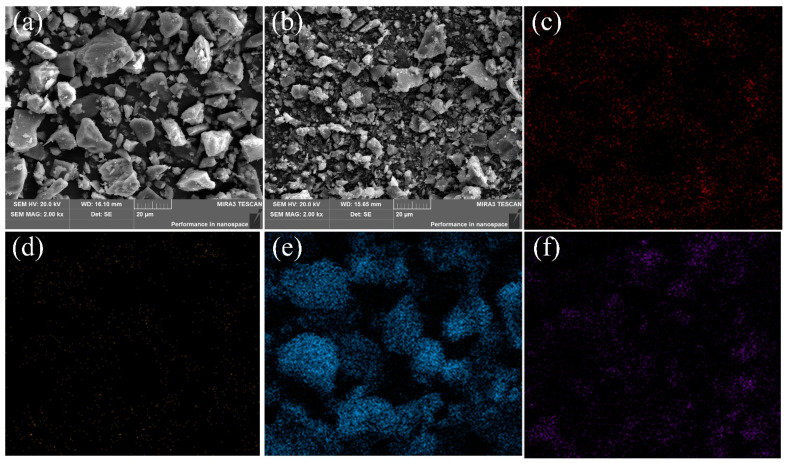
SEM and EDS; (**a**) SEM of TTF; (**b**) SEM of PPY-TTF; (**c**) carbon; (**d**) nitrogen; (**e**) silicon; (**f**) ferrum.

**Figure 4 polymers-14-01540-f004:**
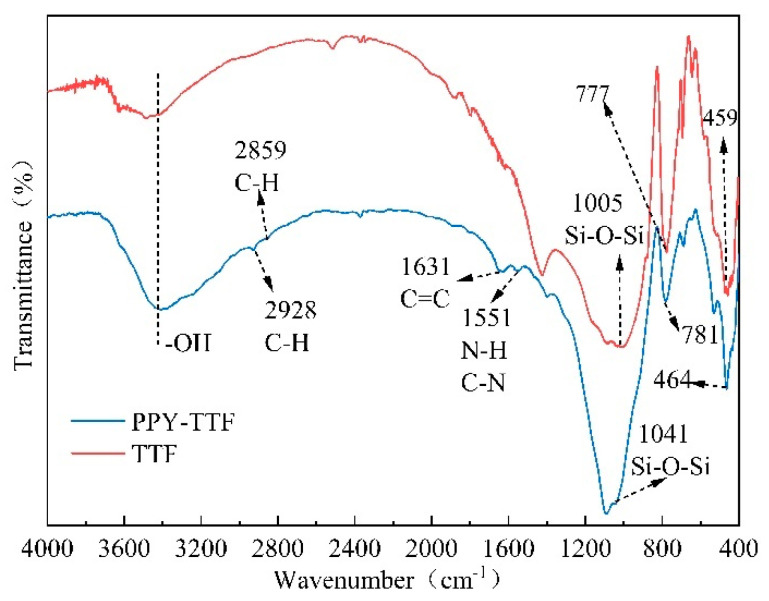
FTIR spectra of TTF and PPY-TTF.

**Figure 5 polymers-14-01540-f005:**
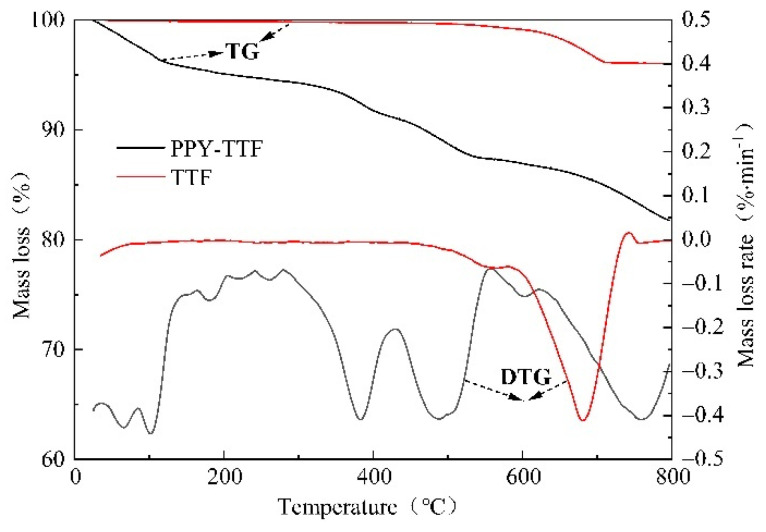
TG and DTG curves of TTF and PPY-TTF.

**Figure 6 polymers-14-01540-f006:**
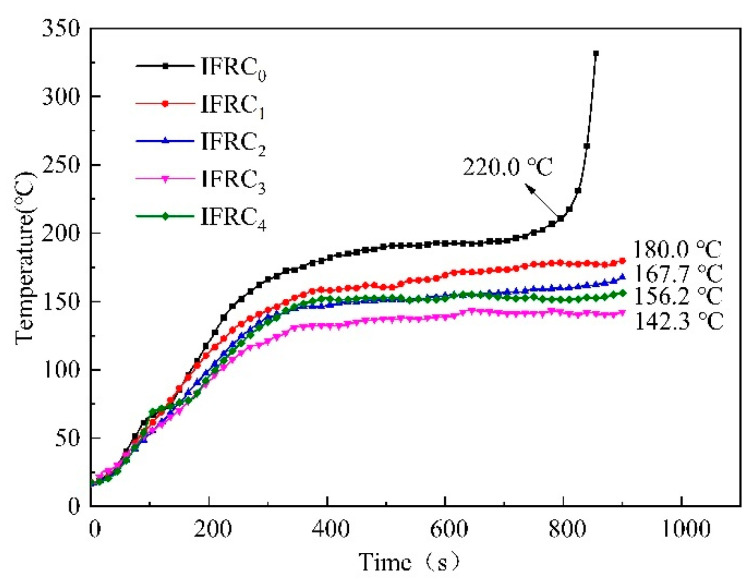
The backside-temperature curves of the substrates coated with IFRC_0_–IFRC_4_ coatings.

**Figure 7 polymers-14-01540-f007:**
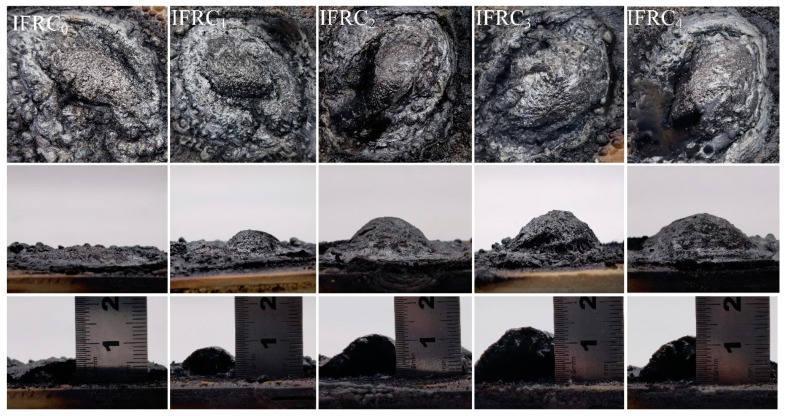
Digital photographs of the obtained char layers from IFRC_0_–IFRC_4_.

**Figure 8 polymers-14-01540-f008:**
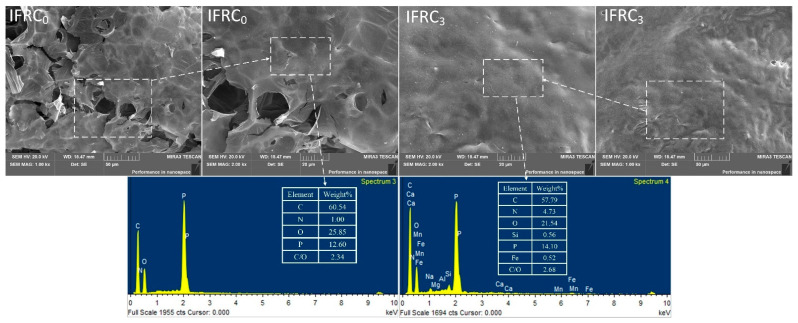
SEM-EDS maps of IFRC_0_ and IFRC_3_.

**Figure 9 polymers-14-01540-f009:**
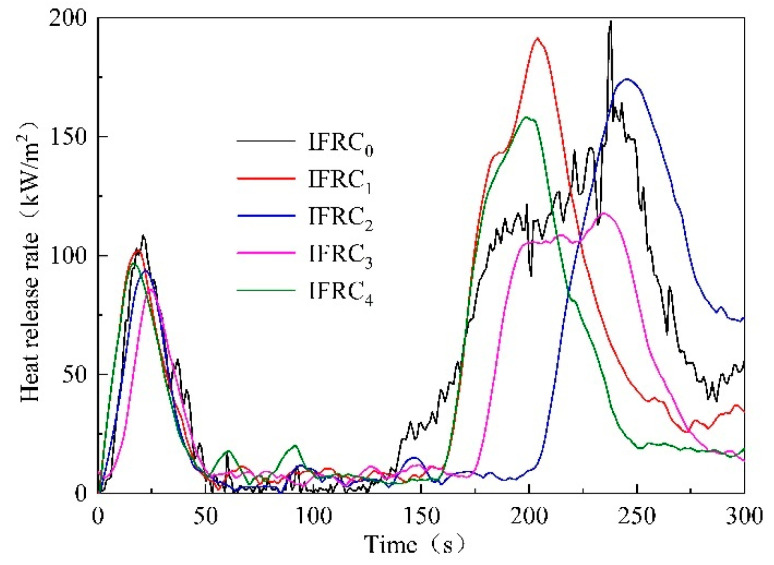
HRR curves of IFRC_0_–IFRC_4_ samples.

**Figure 10 polymers-14-01540-f010:**
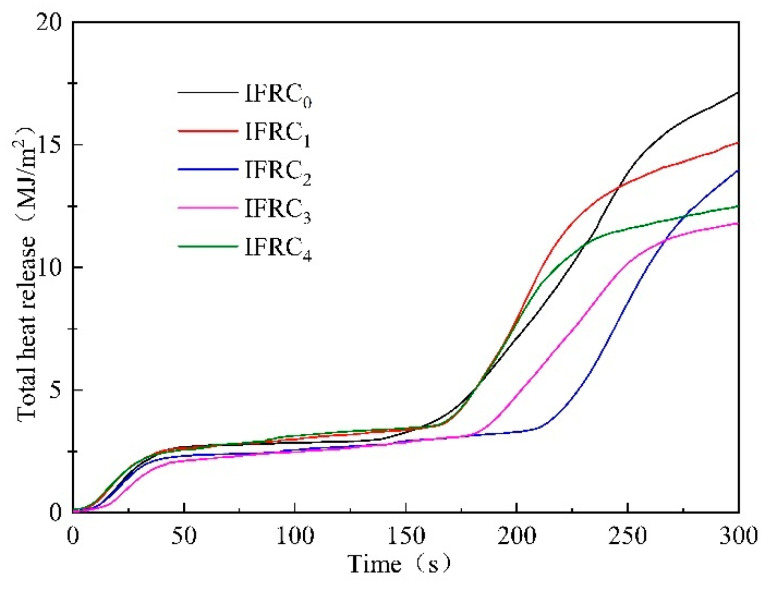
THR curves of IFRC_0_–IFRC_4_ samples.

**Figure 11 polymers-14-01540-f011:**
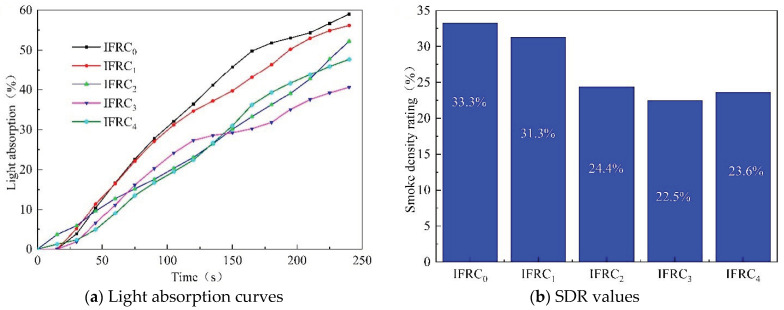
Light absorption curves and SDR values of IFRC_0_–IFRC_4_ samples.

**Figure 12 polymers-14-01540-f012:**
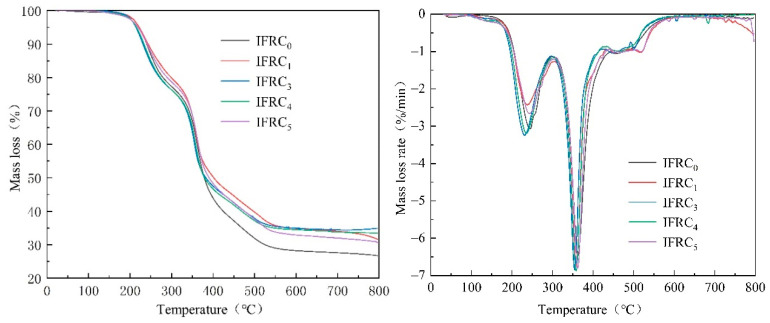
TG and DTG curves of intumescent fire-retardant coatings.

**Figure 13 polymers-14-01540-f013:**
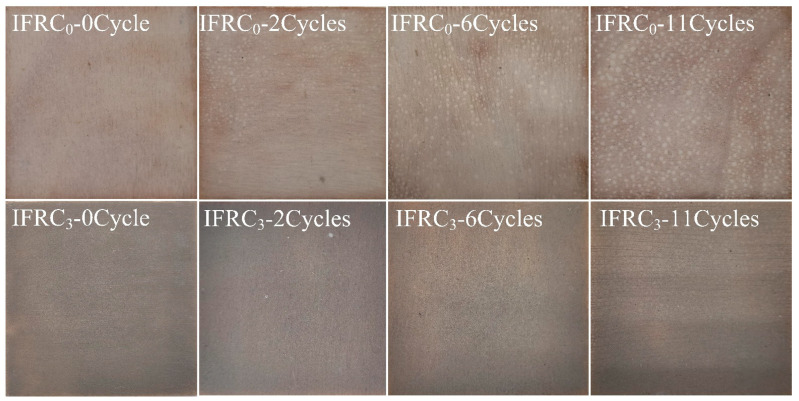
Digital photos of IFRC_0_ and IFRC_3_ after different ageing cycles.

**Figure 14 polymers-14-01540-f014:**
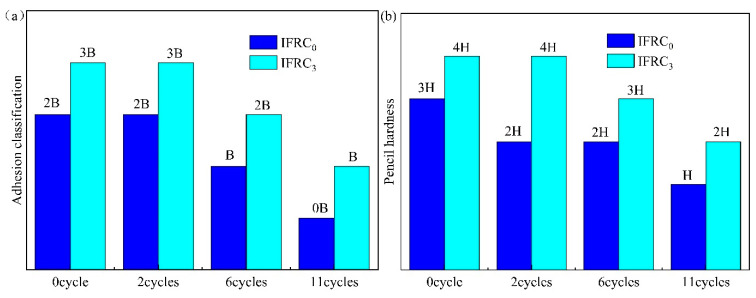
Adhesion classification (**a**) and pencil hardness (**b**) of IFRC_0_ and IFRC_3_ after different ageing cycles.

**Figure 15 polymers-14-01540-f015:**
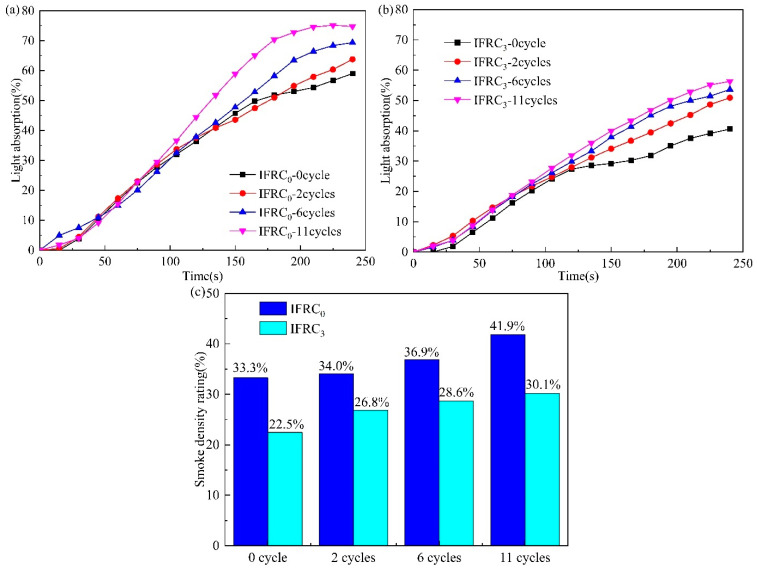
Smoke density testing results after different ageing cycles; (**a**) light absorption of IFRC_0_; (**b**) light absorption of IFRC_3_; (**c**) SDR of IFRC_0_ and IFRC_3_.

**Figure 16 polymers-14-01540-f016:**
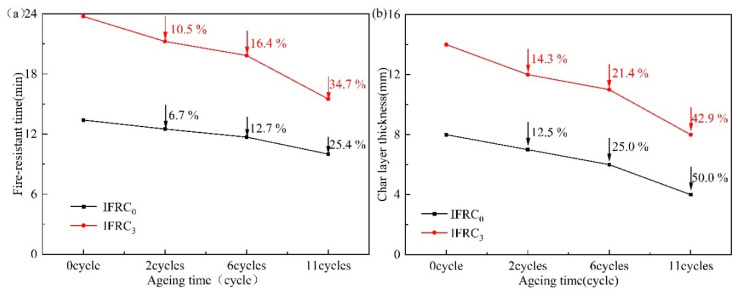
Fire-resistant time (**a**) and char layer thickness (**b**) of IFRC_0_ and IFRC_3_ after accelerated ageing test.

**Figure 17 polymers-14-01540-f017:**
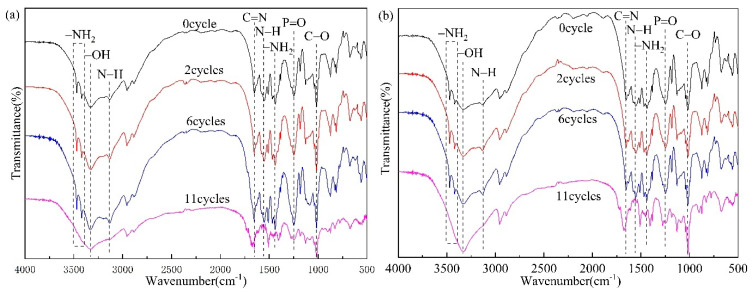
FTIR spectra of IFRC_0_ (**a**) and IFRC_3_ (**b**) after different ageing cycles.

**Figure 18 polymers-14-01540-f018:**
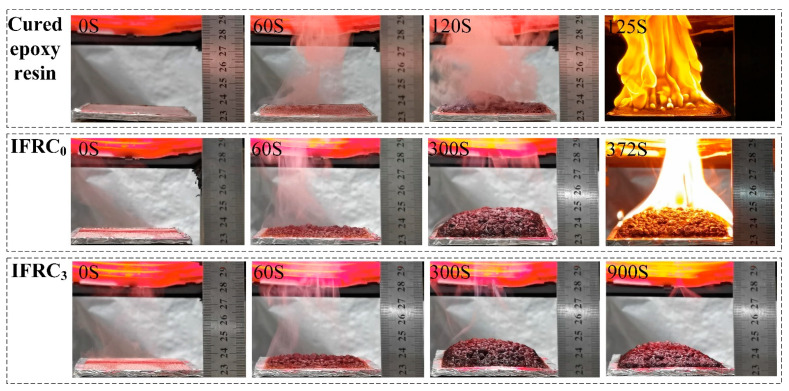
Digital photographs of the cured epoxy resin, IFRC_0_ and IFRC_3_ at expansion test.

**Figure 19 polymers-14-01540-f019:**
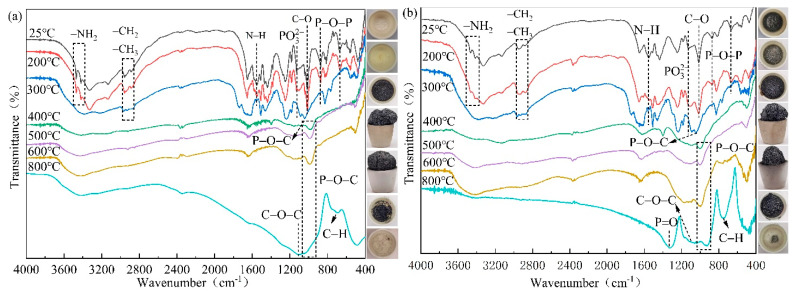
FTIR spectrum and digital pictures of IFRC_0_ (**a**) and IFRC_3_ (**b**) after different treating temperatures.

**Table 1 polymers-14-01540-t001:** Compositions of the intumescent fire-retardant coatings %.

Samples	IFR	TTF	PPY-TTF	Waterborne Epoxy Resin	Defoamer	Dispersant	Waterborne Epoxy Hardener
IFRC_0_	55	0	0	40	0.5	0.5	4
IFRC_1_	54	0	1	40	0.5	0.5	4
IFRC_2_	53	0	2	40	0.5	0.5	4
IFRC_3_	52	0	3	40	0.5	0.5	4
IFRC_4_	50	0	5	40	0.5	0.5	4
IFRC_5_	52	3	0	40	0.5	0.5	4

**Table 2 polymers-14-01540-t002:** XRF test of TTF.

Components	SiO_2_	Al_2_O_3_	CaO	MnO	Fe_2_O_3_	MgO	Na_2_O	K_2_O	WO_3_	SO_3_
Content/%	60.2	10.4	11.8	2.4	6.7	1.6	0.8	1.8	0.4	2.8

**Table 3 polymers-14-01540-t003:** Fire protection performance of intumescent fire-retardant coatings.

Samples	IFRC_0_	IFRC_1_	IFRC_2_	IFRC_3_	IFRC_4_	IFRC_5_
Mass loss/g	3.7 ± 0.1	3.4 ± 0.1	3.0 ± 0.1	2.5 ± 0.1	2.8 ± 0.1	2.8 ± 0.1
Charring index/cm^3^	34.9 ± 0.6	29.9 ± 0.8	23.8 ± 2.4	18.6 ± 0.8	19.6 ± 0.4	18.1 ± 0.4
Flame-spread rating	20.5 ± 1.7	15.8 ± 2.9	8.2 ± 0.8	5.3 ± 2.9	8.2 ± 0.8	8.2 ± 0.3
Intumescent factor	25.0 ± 4.1	31.7 ± 2.4	37.5 ± 2.4	45.0 ± 4.1	42.5 ± 4.1	41.7 ± 2.4

**Table 4 polymers-14-01540-t004:** The relevant thermal decomposition parameters of intumescent fire-retardant coatings.

Samples	*T*_0_/ °C	*T*_m_/ °C	PMLR/ (%/min)	*W*_exp_ (800 °C)/%	*W*_theo_ (800 °C)/%	Δ*W* (800 °C)/%
IFRC_0_	223.1	362.9	6.5	26.7	16.8	9.5
IFRC_1_	222.2	359.1	6.7	31.6	17.4	14.2
IFRC_3_	214.3	354.7	6.8	34.9	19.0	15.9
IFRC_4_	215.8	358.1	6.9	33.5	20.6	12.9
IFRC_5_	219.3	362.0	6.8	30.6	19.4	11.2

Notes: T_0_, the temperature of initial decomposition with the mass loss of 5%; PMLR, peak mass loss rate; T_m_, the temperature of PMLR; *W*_exp_, the experimental char residue; Δ*W* = *W*_exp_ − *W*_theo_; *W*_exp_ of the cured epoxy resin, IFR, TTF, PPY-TTF at 800 °C were 6.7%, 25.2%, 96.0%, 81.8%, respectively.

**Table 5 polymers-14-01540-t005:** FTIR assignments for functional groups of IFRC_0_ and IFRC_3_ after different ageing cycles.

FTIR Band (cm^−1^)	Functional Groups	Observations	
		Intensity	Changes
1439, 3419, 3470	–NH_2_ stretching	Strong	Disappeared
1645	C=N stretching	Weak	Slightly decreased
1552, 3134	N–H stretching	Strong	Disappeared
1249	P=O stretching	Weak	Slightly decreased
1016	C–O stretching	Strong	Significantly decreased

**Table 6 polymers-14-01540-t006:** FTIR assignments for the functional groups of the IFRC_0_ and IFRC_3_ after different treatment temperatures.

FTIR Band (cm^−1^)	Functional Groups	Observations	
		Intensity	Changes
3469, 3419	–NH_2_ stretching	Strong	Disappeared
2956, 2885	–CH_2_ stretching	Strong	Disappeared
1552	N–H stretching	Strong	Disappeared
1324	P=O stretching	Strong	New, increased
1139	C–O–C stretching	Strong	New, increased
1129	PO_3_^2−^ stretching	Strong	Disappeared
1043, 921	P–O–C stretching	Strong	New, increased
1015	C–O stretching	Strong	Disappeared
739	C–H deformation for benzene ring	Strong	New, increased
669, 873	P–O–P stretching	Strong	Disappeared

## Data Availability

The data presented in this study are available upon request from the corresponding author.

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
