# Peer review of "Fabrication of Polypyrrole-Decorated Tungsten Tailing Particles for Reinforcing Flame Retardancy and Ageing Resistance of Intumescent Fire-Resistant Coatings"

_polymers, 2022, doi:10.3390/polym14081540_

Round 1

Reviewer 1 Report

The paper „Fabrication of polypyrrole-decorated tungsten tailing particles…” by F. Wang et al. is well written and describes properties of novel intumescent fire coatings with a substantial numer of analytical tools. There are only minor remarks to improve the paper further:

  • The abstract itself should be self-explaining for the reader without reading the paper to identify abbreviations. Therefore, IFRC should be explained.
  • Abbreviations like APP MEL or PER should be explained, as these are not understood by a non-expert reader
  • As there are many epoxy resins available, at least a general imformation on the chemical basis of the epoxide and the hardener should be given
  • Figure 3, the pictures should be improved as (c), (d), (f) don´t show anything, same for Figure 13, there is no difference between the individual pictures as shown, either the resolution and quality is improved or it is useless
  • What strikes me most technically is the thermogravimetric loss until 100 °C and the next step until 200 °C in the coatings. It shows that the preparation of the coatings had some deficits as under usual conditions there are no water residues left. In addition monomeric Pyrrol has a boiling point of 131 °C, therefore, oligomers should not be volatile below 200 °C and the explanation given is questionable. These facts should be adressed.

Reviewer 2 Report

This paper deals with an interesting topic. However, it should be extensively revised before being published.

  1. The Authors should explain the meaning of all the acronyms to make the text fully readable.
  2. The experimental part should be completed by explaining all reactions and structural characterization of all materials used. In particular, the scheme of Figure 2 is not clear enough.
  3. The recipes should be clear and precise enough to permit reproduction. In section 2.3, in particular, all APP, PER and other compounds should be mentioned explicitly using their chemical name and reporting structures in clear. What are the waterborne epoxy defoamer, dispersant, waterborne, epoxy hardener etc?
  4. The fire protection tests should be described in detail. The same be done for the pencil hardness test.
  5. As a last remark, without all the above information it is just impossible to follow the discussion.

Round 2

Reviewer 2 Report

The Authors have satisfactorily replied to the queries and criticism.  I think that in its present version, this paper can be published.
